# Overcoming Steroid Resistance in Pediatric Acute Lymphoblastic Leukemia—The State-of-the-Art Knowledge and Future Prospects

**DOI:** 10.3390/ijms23073795

**Published:** 2022-03-30

**Authors:** Kamil Kośmider, Katarzyna Karska, Agata Kozakiewicz, Monika Lejman, Joanna Zawitkowska

**Affiliations:** 1Student Scientific Society, Laboratory of Genetic Diagnostics, Medical University of Lublin, Gębali 6, 20-093 Lublin, Poland; kamilkosmider96@gmail.com (K.K.); agatha096@gmail.com (A.K.); 2Department of Pediatric Hematology, Oncology and Transplantology, Medical University of Lublin, Gębali 6, 20-093 Lublin, Poland; katarzynaannakarska@gmail.com; 3Laboratory of Genetic Diagnostics, Medical University of Lublin, Gębali 6, 20-093 Lublin, Poland; monikalejman@umlub.pl

**Keywords:** acute lymphoblastic leukemia, steroids, glucocorticoids, poor prednisone response, IL-7, MAPK/ERK pathway, JAK/STAT pathway, Ras pathway, BH3 mimetics, proteasome inhibitors

## Abstract

Acute lymphoblastic leukemia (ALL) is the most common malignancy among children. Despite the enormous progress in ALL therapy, resulting in achieving a 5-year survival rate of up to 90%, the ambitious goal of reaching a 100% survival rate is still being pursued. A typical ALL treatment includes three phases: remission induction and consolidation and maintenance, preceded by a prednisone prephase. Poor prednisone response (PPR) is defined as the presence of ≥1.0 × 10^9^ blasts/L in the peripheral blood on day eight of therapy and results in significantly frequent relapses and worse outcomes. Hence, identifying risk factors of steroid resistance and finding methods of overcoming that resistance may significantly improve patients’ outcomes. A mitogen-activated protein kinase/extracellular signal-regulated kinase (MAPK-ERK) pathway seems to be a particularly attractive target, as its activation leads to steroid resistance via a phosphorylating Bcl-2-interacting mediator of cell death (BIM), which is crucial in the steroid-induced cell death. Several mutations causing activation of MAPK-ERK were discovered, notably the interleukin-7 receptor (IL-7R) pathway mutations in T-cell ALL and rat sarcoma virus (Ras) pathway mutations in precursor B-cell ALL. MAPK-ERK pathway inhibitors were demonstrated to enhance the results of dexamethasone therapy in preclinical ALL studies. This report summarizes steroids’ mechanism of action, resistance to treatment, and prospects of steroids therapy in pediatric ALL.

## 1. Introduction

Acute lymphoblastic leukemia (ALL) is by far the most common childhood cancer, accounting for 25% of all malignances and 75% of leukemias among children and adolescents [1]. ALL is characterized by abnormal lymphoid cell growth caused by a variety of factors, such as chromosomal abnormalities, transcription factor alterations, and/or chromosomal aneuploidy [2]. The ALL lineage divides the disease into two broad, clinically and physiologically significant categories: precursor B-cell ALL (pre-B ALL) and precursor T-cell ALL (T-ALL). However, numerous new ALL subtypes are currently being found and examined. Over the last few decades, remarkable progress has been made in the treatment of ALL. The Associazione Italiana di Ematologia Oncologia Pediatrica-Berlin-Frankfurt-Münster Acute Lymphoblastic Leukemia (AIEOP-BFM ALL) 2000 study found that the 7-year survival estimates (SE) for 1855 BFM patients were 92.8% [3,4].

Glucocorticoids, such as prednisone or dexamethasone, are essential components of chemotherapy regimens for ALL.

The substantial link between primary glucocorticoid resistance and poor prognosis in ALL emphasizes the significance of glucocorticoid therapy.

In the AIEOP-BFM ALL 2000 protocols, the absolute blast count in the peripheral blood on day eight is critical for classifications. These are (1) Prednisolone good response (PGR): absolute blast count in peripheral blood of <1000/µL; (2) Prednisolone poor response (PPR): absolute blast count in peripheral blood of ≥1000/µL [5,6].

PPR is significantly linked to an increased risk of relapse and unfavorable clinical patients’ outcomes; hence, glucocorticoid resistance poses a significant challenge in the treatment of ALL [7]. However, the precise mechanisms of this resistance have yet to be fully understood. Glucocorticoid treatment may consequently put leukemic cells under selection pressure to acquire genetic changes that diminish a functional steroid response, leading to therapy failure and relapse. On the other hand, subclones harboring mutations responsible for glucocorticoid resistance can be present at the time of diagnosis. Thus, eliminating glucocorticoid-sensitive cells causes the resistant subpopulation to become a dominant clone [8].

Given the significantly worse outcomes in PPR patients, there is a need to develop efficient methods of augmenting the response to glucocorticoids and overcome resistance to the steroid’s treatment. This report summarizes glucocorticoids’ mechanism of action, resistance to treatment, and prospects of steroids therapy in the pediatric ALL.

## 2. Concepts of Glucocorticoid Resistance in Pediatric ALL

### 2.1. Activation of Glucocorticoid Receptor—First Things First

Glucocorticoid receptor (GR, NR3C1) is a protein that is widely expressed and binds glucocorticoid hormones to mediate cellular and tissue-specific effects in development, metabolism, and immune response [9,10]. The human GR is encoded by the *NR3C1* gene (*locus* 5q31.3) and comprises 9 exons (GR protein is encoded by exons 2–9). Several NR3C1 isoforms are produced as a result of alternative splicing events and translation initiation. The primary splice variants of GR that result from alternative splicing of exon 9 are GRα (777 AA) and GRβ (724 AA). Because of its shorter ligand-binding domain, GRβ is unable to bind glucocorticoids and is thought to be an inhibitor of GRα by competing with GRα at the DNA-binding site, and for coregulators. The GRβ isoform contributes to glucocorticoid resistance in the ALL treatment. This resistance can be produced by its antagonism towards GR, as well as the transcriptome changes caused by its presence [11,12,13]. Webster et al. found and demonstrated the function of a tumor-necrosis-factor (TNF)-responsive NF-kB DNA: the consensus-binding element located in the 59-flanking regions of the GR promoter. NF-kB DNA binds to site 5′ of the GR promoter, causing levels of the GRβ protein isoform to be more stable than the GRα protein isoform, making GRβ the dominant endogenous receptor isoform [14]. Levels of TNF-alpha and interleukin 6 (IL-6) were significantly higher in ALL patients at the baseline and before therapy, compared with controls and later study time points, indicating the presence of an inflammatory state in these patients [15]. This could exacerbate GRβ and contribute to the development of steroid resistance. Other GR splice variants, GRγ, GR-A, and GR-P, were discovered to change GR sensibility. GRγ acts as a transcription factor, although it only has 50% of the activity of GRα for conventional glucocorticoid target genes (GTGs). GRγ expression has been linked to resistance to dexamethasone treatment in ALL [13]. Beger et al. established target selectivity of GRγ amplification using sequence-specific primers. Using GRγ-specific amplification in comparison to GR-total (all isoforms) expression in leukemic blasts from patients with either a PGR or a PPR in vivo, researchers discovered that relative GRγ expression was lower in cells from PGR patients compared to PPR patients. These findings were linked to cell survival, with PGR patients’ cells showing a greater activation of apoptosis than PPR patients’ cells [16].

Furthermore, lower overall GR expression may play an important role in the resistance to glucocorticoid treatment. Decreased NR3C1 expression is linked to a poor prognosis and tumor development, and may influence the steroid response [7]. Kuster et al. found that *NR3C1* deletions are prevalent in patients with the *ETV6::RUNX1* fusion gene. *NR3C1*-deleted subjects with minimal residual disease (MRD) had insufficient blast cell clearance in the bone marrow after two months of therapy. Although it is uncertain when the NR3C1 deletions arose, the fact that they were discovered only during relapse does not rule out the potential that they existed in a cell fraction previously [17]. Other studies have also confirmed that *NR3C1* deletions are frequent in the *ETV6::RUNX1*-positive pre-B ALL [18,19,20,21]. A recent study by Liu et al., involving 333 newly diagnosed and 18 relapsed ALL cases, found *NR3C1* mutations in 3 (0.9%) of the newly diagnosed patients and in 2 (11.1%) of the relapsed participants. Four out of five samples with *NR3C1* alterations had loss of function mutations (p. R477H, p. Y478C, p. P530fs, and p. H726P). Next, it was found that the loss of function mutations contributes to resistance to glucocorticoid treatment during in vitro assessments. *NR3C1* mutations caused upregulation of anti-apoptotic members of the B-cell lymphoma 2 (Bcl-2) family and downregulation of the pro-apoptotic Bcl-2 proteins (view Section 2.2) [22]. As reported by van der Zwet et al., 7% of juvenile T-ALL patients had recurrent inactivating *NR3C1* aberrations, such as deletions, missense, and nonsense mutations at the time of diagnosis. However, relative *NR3C1* messenger RNA (mRNA) expression in primary diagnostic patient samples did not correlate with steroid response [8].

Transfection of Reh (pre-B ALL cells), and Jurkat (T-ALL cells) cell lines with a pLentiC-Myc-DDK-NR3C1 lentiviral vector, carrying wild-type *NR3C1*, forces NR3C1 expression in dexamethasone-resistant Reh (pre-B ALL cells), and Jurkat (T-ALL cells) cell lines. This ultimately caused significant reduction of proliferation rate and increase in apoptosis, as compared to Reh and Jurkat cells transfected with an empty control vector. This demonstrates the association of the NR3C1 expression and glucocorticoid resistance in vitro [23].

### 2.2. Bcl-2 Protein Family as a Critical Mediator of Glucocorticoid-Induced Apoptosis

Nevertheless, effective activation of GR is only a partial success, as these receptors exert multiple effects in leukemic cells. Glucocorticoids bind to the GR in the cytoplasm, resulting in NR3C1 homodimerization. These homodimers are subsequently transferred to the nucleus, where they regulate GTGs expressions by binding with specific DNA sequences, called glucocorticoid response elements (GREs) [24,25]. In most ALL cases, GR activation leads to the apoptosis of leukemic cells and consequently reduces the blasts count. Unfortunately, steroid-resistant clones can be occasionally encountered, significantly worsening the prognosis [26]. Therefore, in recent years, cellular consequences of GR activation and their malfunction were the subject of intensive research. Although GR has been demonstrated to regulate a multitude of genes, this review focuses on the most important signaling pathways affected by GR in the context of ALL treatment.

The Bcl-2 protein family is a group of proteins sharing Bcl-2 homology (BH1, BH2, BH3, and BH4) domains and taking part in the regulation of apoptosis. Among proteins promoting apoptosis are the BH3-only proteins, transmitting signals from different pathways, the inter alia Bcl-2-interacting mediator of cell death (BIM), the BH3-interacting-domain death agonist (Bid), the p53-upregulated modulator of apoptosis (Puma), the Bcl-2-associated agonist of cell death (Bad), phorbol-12-myristate-13-acetate-induced protein 1 (Noxa), and activator of apoptosis harakiri (Hrk). The critical mediators Bcl-2-associated X-protein 4 (Bax) and Bcl-2 homologous antagonist/killer (Bak) initiate apoptosis by causing permeabilization of the outer mitochondrial membrane. Bax and Bak activation is suppressed by the anti-apoptotic members of the BCL-2 protein family, such as Bcl-2, B-cell lymphoma extra-large (Bcl-XL), B-cell lymphoma-w (Bcl-w), and the induced myeloid leukemia cell differentiation protein (Mcl-1). Suppression of Bax and Bak synthesis is overcome when the BH3-only proteins bind to the anti-apoptotic members of the Bcl-2 family, causing caseation of Bax and Bak blockage [27]. However, the results of several studies indicate that the BH3-only proteins may also be able to initiate apoptosis via direct interaction with Bak or Bax [28,29,30,31,32,33,34,35].

Interestingly, anti-apoptotic proteins are basally overexpressed in ALL, whereas pro-apoptotic members of the Bcl-2 family (for instance, Bax) are shown to be downregulated, pushing the balance between pro and anti-apoptotic factors towards the cell’s survival, which significantly contributes to treatment resistance [36,37,38,39,40,41,42,43]. The Bcl-2 protein family has been demonstrated to be regulated via GR activation. One of the most important pro-apoptotic proteins involved in steroid-induced apoptosis of leukemic cells is BIM, encoded by the *BCL2L11* gene, located at 2q13 [44,45,46,47,48,49]. Wang et al. found that BIM was significantly induced after dexamethasone treatment of glucocorticoid-sensitive T-cell lymphoma cell lines (S49.A2 and WEHI7.2). It has been observed that, in the case of inhibited transcription or protein synthesis, BIM level drops to a level that is difficult to detect. Thus, it was concluded that dexamethasone treatment results in de novo transcription and translation of *BCL2L11*. Furthermore, BIM was also induced in glucocorticoid-sensitive T-ALL line CEM-C7 after dexamethasone treatment [46]. Zhao and colleagues also detected that BIM was up-regulated in CEM-C7 cells after their incubation with dexamethasone, which caused dose-dependent and time-dependent apoptosis of those cells. However, when glucocorticoid-resistant CEM-C1 cells were incubated with dexamethasone, not only did it not cause apoptosis, but it also did not manage to significantly induce BIM [45]. Erlacher et al. described that 20 h after a single intraperitoneal dexamethasone injection in wild-type mice, the reduction of thymic immature CD4+8+ double-positive cells count, as well as a decrease of pre-B-cell numbers in the bone marrow, occurred. However, when dexamethasone was administered in the BIM-deficient or Puma-deficient mice, the observed reductions of thymocytes and immature B-cells were significantly lower, indicating that pro-apoptotic members of the Bcl-2 family BIM and Puma are critical in the glucocorticoid-induced apoptosis of ALL cells [47]. Schmidt et al. obtained peripheral blood lymphocytes of 13 glucocorticoid-sensitive children (3 T-ALL, 10 pre-B ALL) prior to prednisolone therapy initiation and at 6- to 8-h intervals after the beginning of glucocorticoid treatment. *BCL2L11* was up-regulated in two of the T-ALL patients and in four of the pre-B ALL children. This indicates that although BIM induction is the most recognizable mechanism of glucocorticoid related apoptosis in ALL, there is a considerable group of glucocorticoid-sensitive patients, in which apoptosis is initiated by different actions [49].

Another important protein involved in GR-induced apoptosis is the anti-apoptotic Bcl-2, encoded by the *Bcl-2* gene (18q21.3); its role in apoptosis was discovered as early as 1988 [50]. Laane et al. performed an analysis of Bcl-2 proteins’ expression in cells obtained from the bone marrow of 12 children (9 with pre-B ALL, 3 suffering from T-ALL), and in the pre-B ALL cell lines RS4 and Reh, as well as in T-ALL line CCRF-CEM. Cell cultures were divided into the untreated control and dexamethasone-treated groups and incubated for 72 h. It turned out that in the highly sensitive RS4 line, Bcl-2 was significantly suppressed. This down-regulation of *Bcl-2* correlated with apoptosis. Furthermore, Bcl-XL (encoded by *BCL2L1* gene) was also significantly suppressed. Importantly, when apoptosis was induced by doxorubicin, no changes in the *Bcl-2* or *BCL2L1* expression occurred, indicating that Bcl-2 protein family is critically important, specifically in the glucocorticoid-induced apoptosis of ALL cells. Suppression of Bcl-2 and Bcl-XL was also found in the CCRF-CEM cells, whereas in the resistant Reh cells, levels of the proteins were not changed [51]. In 2015, Jing et al. published the results of their study on the effects of dexamethasone treatment of pre-B ALL patient-derived xenografts. Microarray analysis of gene expression in 10 xenografts (5 glucocorticoid-sensitive and 5 glucocorticoid-resistant) after dexamethasone administration into the engrafted mouse revealed up-regulation of *BCL2L11* in the sensitive cells, whereas in the resistant xenografts, *BCL2L11* expression was distinctly lower. It is noteworthy that *Bcl-2* was expressed exactly opposite. Interestingly, there was a significant correlation between *Bcl-2* down-regulation and BIM induction (*p* < 0.05), indicating the significant role of coordinated *Bcl-2* and *BCL2L11* regulation in the glucocorticoid-induced apoptosis of leukemic cells. Further analysis led to the discovery of novel *BIM* IGR (intronic GR binding region), at which GR binding was detected following dexamethasone administration in vivo. Moreover, two GREs were identified within *BIM* IGR, and GR’s binding to these elements was detected, thus revealing the direct influence of dexamethasone on *BIM* expression [52].

### 2.3. Role of Proteasomal Degradation in Resistance to Glucocorticoid Treatment in Pediatric ALL

F-Box and WD Repeat Domain-Containing 7 (FBXW7) inactivation was found to be associated with PGR and overall better outcomes in the ALL by numerous studies [53,54,55,56,57,58,59]. According to Malyukova et al.’s investigation, in a glycogen-synthase-kinase-3 (GSK3)-dependent mechanism, FBXW7 mediates ubiquitylation and proteasomal degradation of GR. Inactivation of FBXW7 increases GRα stability and activity, boosting transcription of GTGs, including pro-apoptotic genes. GSK3-mediated phosphorylation of GRα, at S404, is needed for FBXW7 binding to GRα, which eventually targets it for proteasomal degradation. The S404A GRα mutant was unable to undergo FBXW7-mediated ubiquitination, limiting its proteasome destruction. Hence, FBXW7 activity and expression have implications for glucocorticoid sensitivity through modulating glucocorticoid-mediated declines in the GR pool [60,61].

Dexamethasone significantly (*p* < 0.05) decreased pro-apoptotic Noxa levels in the T-ALL cell lines CEM-C7H2 and Jurkat^GR^, as well as in the pre-B ALL line 697/EU-3. Further, glucocorticoid treatment repressed *Noxa* mRNA as demonstrated by real-time polymerase chain reaction (RT-PCR) analysis. Dexamethasone treatment of derivatives of CEM-C7H2-2C8 cells, in which *Noxa* expression can be triggered by doxycycline but cannot be modified by glucocorticoids, revealed regression of doxycycline-induced Noxa levels. Further, when these cells were cultured with proteasomal inhibitors (MG-132 or lactacystin) and incubated with dexamethasone, it resulted in an increased Noxa level, suggesting that glucocorticoids cause Noxa proteolysis via the proteasome. Moreover, in CEM-C7H2-2C8 cells modified to express BIM_EL_, after doxycycline administration, apoptosis was induced faster by doxycycline than by following the exposure to dexamethasone with end BIM_EL_ levels similar in both groups, which may be explained by the fact that glucocorticoids induce both pro- and anti-apoptotic signaling. When the glucocorticoids could not decrease *Noxa* expression in the modified CEM-C7H2-2C8 cells, the dexamethasone-induced apoptosis was significantly enhanced in comparison to the unaltered CEM-C7H2-2C8 treated with dexamethasone. Importantly, cell death was increased to a similar level as in the group in which BIM_EL_ was triggered by doxycycline alone. This indicates that glucocorticoids’ anti-apoptotic action exerted via proteasomal degradation of Noxa significantly weakens its pro-apoptotic effect on the leukemic cells [62]. Thus, proteasomal inhibitors might be beneficial in overcoming glucocorticoid resistance.

### 2.4. IKZF1 Alterations and Glucocorticoid Resistance

DNA-binding protein Ikaros is a transcriptional regulator belonging to the zinc finger protein family, encoded by the *IKZF1* gene. Ikaros has been found to bind to target DNA sequences with a specific TGGGAA consensual motif, and subsequently regulate the gene’s transcription. The exact mechanism in which Ikaros exerts its action has not yet been found, however, it was described that it can trigger chromatin remodeling. Importantly, Ikaros was found to regulate the development of all lymphoid lineages and alterations in the *IKZF1* gene, leading to the development of various lymphological malignancies [63,64,65,66]. An important subgroup, *IKZF1*^plus^, consists of patients with *IKZF1* deletions, combined with deletions in *CDKN2A* or *CDKN2B* (only homozygous deletions), or the *PAX5* or PAR1 region (*P2RY8-CRLF2*), in the absence of *ERG* deletion [67].

Marke et al. demonstrated that splenic B-cells obtained from *IKZF1*^+/−^ mice are less prone to apoptosis after incubation for 48 h with prednisolone or dexamethasone than wild-type B-cells (*p* < 0.001). Furthermore, when *IKZF1* was silenced in glucocorticoid-sensitive RS4;11 and NALM6 pre-B ALL cell lines, a significant glucocorticoid resistance was induced, which correlated with alterations in GTG’s expression. The 152 *IKZF1*-wild-type pre-B ALL samples and 37 pre-B ALL samples with mutation/deletion of the *IKZF1* gene were incubated with prednisolone and dexamethasone. The median lethal concentration (LC_50_) after 96 h in samples with *IKZF1* alteration was 10-fold higher for prednisolone (*p* = 0.004) and 20-fold higher for dexamethasone (*p* = 0.0001) than in wild-type samples [68].

Philadelphia chromosome-like (Ph-like) ALL is an ALL subtype characterized by the gene expression profile similar to Philadelphia chromosome-positive ALL, although without the *BCR::ABL1* fusion gene. *IKZF1* mutations are especially prevalent in the Ph-like ALL, as they are found in 68% of such patients. Further, 16% of patients with the *BCR::ABL1*-negative, pre-B ALL patients harbor mutations in the *IKZF1* gene. *IKZF1* malfunction is related to a significantly worse prognosis, both in Ph-like and in *BCR::ABL1*-negative, pre-B ALL patients [69,70,71,72]. The correlation between prednisone response on day eight and *IKZF1*-deletion status in 646 pre-B ALL pediatric patients was examined. It turned out that *IKZF1* deletions were more prevalent in PPR (*n* = 52) patients than in PGR (*n* = 594) patients (27% vs. 14%; *p* = 0.015) [68]. Similar results were obtained during the Japan Association of Childhood Leukemia Study. Patients with pre-B ALL (*n* = 1174) were assigned to SR (standard risk), HR (high risk), and extremely high risk (ER) groups. ER group was characterized by initial prednisone resistance and poor response on day 15. Patients in the ER group had a significantly lower 5-year event-free survival (EFS) rate than SR and HR participants (66.0 ± 6.3% vs. 86.3 ± 2.2%; *p* = 0.0005). *IKZF1* deletions were found in 15 of 71 patients in the ER group and 22 of 261 patients in the remaining groups (21% vs. 8.4%; *p* = 0.003). However, the difference in 5-year EFS, between *IKZF1*-mutated and *IKZF1*-wild-type patients in the ER group, was not statistically significant, indicating that among these patients, *IKZF1* alterations are not the only factor that causes poor prognosis. On the contrary, among patients in the SR and HR groups, the presence of *IKZF1* alteration significantly worsened the 5-year EFS rate (61.2 ± 10.8% in the HR+SR IKZF1-deletion patients vs. 85.9 ± 2.9% in the HR+SR *IKZF1*-wild-type participants; *p* = 0.0005) [73]. Recently, Braun et al. completed a study in which 373 Polish pediatric pre-B ALL patients were treated according to the ALL-IC BFM 2009 protocol. It turned out that patients with the *IKZF1* deletion or *IKZF1*^plus^ pattern are characterized by a higher PPR rate than *IKZF1*-wild-type participants (26.5% vs. 12.5% vs. 7.6%; *p* = 0.010) [74].

## 3. Signaling Pathways Contribute to Glucocorticoid Resistance in Pediatric ALL—Prospects for Future Treatment

### 3.1. Interleukin-7 Signaling Pathway and Glucocorticoid Resistance in ALL

Interleukin-7 (IL-7) plays a crucial role in the T- and B-cell development. It is responsible for survival and regulating the functions of peripheral mature T-cells [75,76,77,78,79]. IL-7 acts on cells via a heterodimer IL-7 receptor (IL-7R) comprised of subunit alpha (IL-7Rα, CD127) and the common γ chain (γc) subunit [79]. Binding to IL-7R results in activation of associated tyrosine Janus kinases (JAK): JAK1 (by IL-7Rα) and JAK3 (by γc) [79,80]. JAK 1 and JAK3 induce phosphorylation of signal transducer and activator of transcription proteins (STAT) transcriptional factors, mainly STAT5A and STAT5B. Furthermore, STAT1 and STAT 3 are also activated. Next, STAT proteins translocate into the nucleus where they regulate cell-growth and survival via inter alia, increasing Bcl-2 expression. STAT5 proteins also activate the PI3K/AKT signaling pathway [79,80,81]. Increased activation of the IL-7 pathway causes leukemogenesis in multiple in vitro studies. Upregulation of IL-7 signaling pathway was observed in the leukemic cells obtained from T-ALL patients, as well as in the T-ALL cell lines, and was associated with increased cell survival and proliferation [81,82,83,84,85,86,87,88]. Furthermore, activation of the IL-7 signaling pathway was showed to be associated with T-ALL resistance to glucocorticoid treatment [89,90,91].

The findings described above are also important for the ALL treatment. Mutations in IL-7R/JAK signaling have been found in pre-B ALL, and thus are inhibitors of this pathway, as well as blockers of downstream signaling, which could be beneficial in pre-B ALL therapy [87,92,93].

Delgado-Martin et al. demonstrated that in vitro IL-7-related glucocorticoid resistance could be overcome by inhibiting the JAK/STAT pathway. IL-7-related resistance was induced by incubating the non-early T-cell progenitor (ETP) (*n* = 22) and ETP T-ALL (*n* = 10) xenograft samples with IL-7. Some of the ETP T-ALL samples were resistant to glucocorticoids irrespective of the IL-7 addition, however, others were resistant only when incubated with IL-7. Similarly, glucocorticoid resistance in non-ETP T-ALL samples was strongly IL-7-dependant (*n* = 10), partially dependent (*n* = 7) and IL-7-independent (*n* = 5). Ruxolitinib, a JAK1/2 inhibitor, in combination with dexamethasone, restored glucocorticoid sensitivity in the IL-7-dependent non-ETP and ETP T-ALL samples. Furthermore, when cells were incubated with the JAK3 inhibitor, a similar effect occurred. Importantly, the augmentation of apoptosis was significantly higher when JAK inhibitors were used in combination with dexamethasone, instead of a single treatment. Ruxolitinib and dexamethasone co-treatment was found to decrease the Bcl-2 level, which was elevated in the IL-7-dependent cells. In contrast, there was no augmentation of Bcl-2 level in IL-7-independent samples. Interestingly, venetoclax (Bcl-2 inhibitor), in combination with dexamethasone, sensitized IL-7-dependent cells to glucocorticoid treatment; however, the effects of this drug combination were far worse than those of the ruxolitinib and dexamethasone combination [90].

In around half of the Ph-like ALL rearrangement, the cytokine receptor-like factor 2 *(CRLF2)* gene is found, which leads to *CRLF2* overexpression. This, in turn, is frequently associated with activation of JAK/STAT cascade [94,95]. Ruxolitinib, in combination with vincristine, dexamethasone, and an L-asparaginase (VXL) induction-type treatment regimen, was found to be efficient in mice engrafted with Ph-like ALL xenografts harboring *JAK* activating mutations. This drug combination acted synergistically and prolonged disease remission [96].

It is worth noting that a phase I clinical study of a monoclonal antibody against IL-7Rα (GSK2618960) has been completed. As indicated by this double-blind study conducted in 18 healthy individuals, GSK2618960 treatment is tolerated and efficiently blocks IL-7 signaling. However, it did not induce any effect on healthy T-cells. Nevertheless, GSK2618960 can still decrease pro-survival cascade activation in leukemic cells; thus, further studies are needed [97]. Furthermore, another monoclonal antibody against IL-7Rα (named B12), which blocks both the wild-type and mutated IL-7Rα, was developed. B12 was found to block IL-7 and mutant IL-7Rα signaling, and to induce apoptosis in vitro. Further, B12 delayed T-ALL progression in vivo in the T-ALL engrafted mice. It was also found to potentialize dexamethasone-induced apoptosis in vitro [98]. Given the good tolerance of IL-7Rα inhibitors during the phase I clinical study, as well as promising results of preclinical research, efficiency of these drugs in ALL treatment should be evaluated as soon as possible.

### 3.2. Activation of PI3K/AKT/mTOR Signaling Cascade Prevents GR from Translocation to the Nucleus

An alternate underlying mechanism of glucocorticoid resistance dependent on NRC3C1 activation could be the NR3C1 inhibitory phosphorylation, which reduces its nuclear localization and transactivation ability to activate critical downstream GTGs. Serine/threonine kinase (AKT) 1 binds to and phosphorylates the NR3C1 protein, inhibiting its nuclear translocation, according to Piovan et al. This finding suggests that activation of AKT1 may play a role in the development of glucocorticoid resistance in ALL. In vitro and in vivo, pharmacological inhibition of AKT with MK2206 significantly restores glucocorticoid-induced NR3C1 translocation to the nucleus, increases the sensitivity of T-ALL cells to the glucocorticoid therapy, and successfully reverses glucocorticoid resistance [99].

AKT1 is a part of phosphatidylinositol 3-kinase (PI3K)/AKT/mammalian target of a rapamycin kinase (mTOR) signaling cascade. In leukemia, the PI3K/AKT/mTOR pathway is usually activated and plays a role in leukemogenesis, especially in T-ALL. Increased cell metabolism, proliferation, and decreased apoptosis are the result of continuous stimulation of this mechanism. Activating mutations in PI3K genes, as well as downstream effectors of the cascade, such as AKT and mTOR, causes overexpression of this pathway. Moreover, inactivating mutations in the *PTEN* gene have been reported in T-ALL patients. PTEN, in turn, is a crucial inhibitor of the PI3K/AKT cascade [100]. Activation of PI3K/AKT signaling has been also linked to the increase in the level of Bcl-2, which is an anti-apoptotic protein [88]. Furthermore, the FBXW7 seems to be regulated by the PI3K/AKT, thus, blocking this pathway may result in increasing the GRα level [101]. Therefore, this may be another mechanism of glucocorticoid resistance induced by the PI3K/AKT axis. However, this requires further laboratory studies, as so far, the influence of PI3K/AKT activation on FBXW7 in leukemic cells has not been clarified.

Wandler et al. used PI3K inhibitor pictilisib (GDC-0941) to treat primary murine, genetically heterogeneous T-ALLs, both in a single treatment or in combination with dexamethasone. Authors reported that 68% of T-ALL mice that relapsed after initially responding to glucocorticoid treatment in vivo had a decreased or missing GR protein expression. Adding GDC-0941 to glucocorticoid treatment resulted in modestly prolonged survival (median 31 versus 40 days; *p* = 0.0805). Importantly, GR protein levels were reduced in 40% of relapsed human T-ALL samples, implying that GR expression loss is a primary source of glucocorticoid resistance. This finding is consistent with clinical studies linking a poor response to glucocorticoid treatment during induction to a higher likelihood of relapse [102]. Pictilisib is demonstrated also by other studies to be efficient in the T-ALL with PI3K/AKT/mTOR pathway activation [103,104,105]. Several other PI3K inhibitors have been efficient in treating ALL during preclinical studies. Idelalisib successfully treated ex vivo pre-B ALL samples with *TCF3::PBX1* gene fusion [106]. Further, idelalisib was also found to be effective ex vivo against T-cell leukemia-lymphoma samples [107]. Buparlisib treatment was beneficial in vitro in the T-ALL cell lines [108,109]. Evangelisti et al. found that PI3K p110 inhibitors (ZSTK-474, AS-605240, CAL-101, and IPI-145) induce apoptosis both in pre-B ALL cell lines, as well as ex vivo in the samples obtained from pre-B ALL patients. Reh6 and Nalm6 cells were incubated with human bone marrow mesenchymal stem cells, which induce glucocorticoid resistance in vitro. P110 inhibitors (ZSTK-474 and IPI-145) were found to increase the effects of dexamethasone therapy in these cell lines, thus overcoming glucocorticoid resistance. Furthermore, PI3K inhibitor (IPI-145 or ZSTK-474) treatment of KOPN8 and Nalm6 reversed the AKT-induced impartment of GR translocation to the nucleus. Dexamethasone, in co-treatment with PI-145 or ZSTK-474, was found to be more beneficial than a single drug treatment against pre-B ALL patient samples [110].

MTOR inhibitors might be beneficial in treating primary human ALL. In their studies, Teachey et al. demonstrated the activity of mTOR inhibitors in preclinical models of ALL [111]. In addition, mTOR inhibitors improve methotrexate sensitivity by downregulating dihydrofolate reductase expression [112]. In particular, the combined inhibition of PI3K and the mTOR complex may provide an effective treatment for acute leukemia. In fact, they had a much stronger cytostatic effect on ALL cells than everolimus, according to Wong et al. [113]. Dactolisib (BEZ235) is an imidazoquinoline derivative that is a potent dual pan-class I PI3K and mTOR inhibitor. It inhibits downstream PI3K effectors in numerous preclinical models, including cell lines and xenografts, resulting in efficient reduction of tumor proliferation and growth [114]. As showed by the phase I clinical study performed by Lang et al., dual inhibition of PI3K and mTOR by dactolisib induces responses in 30% of ALL cases. Dactolisib, at the dose of 400 mg/day, was poorly tolerated and severe toxicity events occurred, especially gastrointestinal (mainly stomatitis). Considering that most dose-limiting toxicities such as fatigue, diarrhea, nausea, and mucositis were noted with both PI3K and mTOR inhibitors, it is not unexpected that pan-PI3K and mTOR inhibition resulted in a high prevalence of adverse events at the dose of 400 mg/day. On the other hand, 300 mg/day was far better tolerated by the patients; therefore, this dose was established as recommended for phase II studies [115]. It is worth noting that dactolisib was found to increase the incidence of grade 3–4 adverse events in evaluable patients in several other clinical studies [116]. Regrettably, currently there is no ongoing clinical study of dactolisib, according to clinicaltrials.gov.

### 3.3. The MAPK-ERK Pathway

Glucocorticoid resistance of T-ALL leukemic cells, related to upregulation of the IL-7 pathway, was mainly associated with STAT5-induced increase in the anti-apoptotic Bcl-2. Furthermore, it was observed that glucocorticoid treatment results in upregulation of IL-7Rα, thus creating a vicious cycle [89,90]. However, recently, additional mechanisms of IL-7-pathway-related resistance have been described.

Li et al. discovered mutations of IL-7 pathway genes in 47 (32%) samples obtained from 146 pediatric T-ALL patients. These mutations affected *IL7R*, *JAK1*, *JAK3*, *NF1*, *NRAS*, *KRAS*, and *AKT* genes. In 28 out of 97 prednisolone-treated samples, the IL-7 pathway mutations were associated with resistance to glucocorticoid treatment (*p* = 0.033). Interestingly, patients with IL-7 pathway mutations were characterized by significantly (*p* = 0.009) worse clinical outcomes, as compared to children without these mutations. Furthermore, those mutations caused activation of the PI3K/AKT pathway, resulting in an increase in Mcl-1 and Bcl-XL levels. Furthermore, higher levels of inactivated GSK3B, which is an important kinase that regulates BIM’s function, were found. Consequently, a higher ratio of phosphorylated/unphosphorylated BIM was observed. These changes were associated with mitogen-activated protein kinase (MAPK)—extracellular signal-regulated kinase (ERK) pathway activation [89]. The MAPK-ERK pathway takes part in controlling cells’ growth, proliferation, survival, and division. Upon MAPK-ERK activation, ERK migrates to the nucleus where it directly phosphorylates target proteins or controls other kinase activity [117]. When glucocorticoid-resistant cell lines were treated with the MAPK inhibitor (CI1040), there was a significant enhancement in GSK3B activation, as well as an increase in the non-phosphorylated BIM level [89].

A recently published study by van der Zwet et al. demonstrated that SUPT-1 cells expressing cysteine mutants IL-7Rα^PILLT240−244RFCPH^, IL-7R-α^PIL240−242QSPSC^, and IL-7Rα^LT243−244LMCP^, exhibited glucocorticoid resistance and activation of the downstream MAPK-ERK. In contrast, in the cells expressing the wild-type and the non-cysteine IL-7Rα mutant, MAPK-ERK upregulation and glucocorticoid resistance were not found. The MAPK-ERK pathway was also upregulated in the glucocorticoid-resistant SUPT-1 cells expressing *JAK1^R724H^, JAK1^T901A^*, *NRAS^WT^*, or *NRAS^G12D^*, whereas in the glucocorticoid-sensitive SUPT-1 cells (expressing wild-type *JAK1*) MAPK-ERK was not activated. It was demonstrated that MAPK-ERK upregulation causes increase in the phosphorylation of BIM_EL_ and BIM_L_ isoforms. Further, it was showed that ERK is responsible for the direct phosphorylation of BIM. BIM’s phosphorylation was found to cause impaired BIM’s binding to Bcl-2, Mcl-1, and Bcl-XL, which prevents their inactivation and, thus, promotes cells’ survival. Furthermore, in the *JAK1^T901A^*-and-*NRAS^G12D^*-expressing SUPT-1 cells, MAPK 1/2 inhibitors selumetinib and trametinib prevented BIM’s phosphorylation in a dose-dependent manner. A similar effect was achieved in SUPT-1 cells expressing JAK1^T901A^ treated with JAK1/JAK2 inhibitor ruxolitinib. The response to the glucocorticoid treatment in the 46 T-ALL patient-derived xenografts with or without IL-7 was assessed. IL-7 addition induced glucocorticoid resistance in 12 (26%) of all xenografts. Interestingly, in those samples, MAPK-ERK signaling was activated by IL-7, indicating that physiological IL-7 signaling may activate the downstream MAPK-ERK pathway in T-ALL. Thus, in T leukemic cells, MAPK-ERK can be upregulated both by mutant and physiological IL-7 signaling [118]. Interestingly, the MAPK-ERK signaling pathway is not activated by the IL-7 signaling in the healthy T-cells [119]. Xenografts, in the absence of IL-7, were treated using MAPK inhibitors (selumetinib, trametinib, and binimetinib) and ruxolitinib. Ruxolitinib did not cause therapeutic effects, in contrast to MAPK inhibitors, which induced cytotoxicity in most of the samples. Six samples with IL-7-induced glucocorticoid resistance were then treated with ruxolitinib. The significant relationship (*p* = 0.0039) between sensitivity to ruxolitinib treatment and IL-7-enhanced cells viability was found. Combined treatment with selumetinib and prednisolone was highly synergic, both in xenografts with IL-7-induced glucocorticoid resistance, and in the samples with not-IL-7-related glucocorticoid resistance. Furthermore, such a synergistic effect was also found in the glucocorticoid-sensitive xenografts. Ruxolitinib and prednisolone co-treatment was efficient only in one xenograft, with not- IL-7-related glucocorticoid resistance in the presence of IL-7, and in one xenograft with IL-7-induced glucocorticoid resistance without IL-7. In both samples, there was a significant STAT5 overexpression, which explains ruxolitinib efficiency. Therefore, ruxolitinib has limited clinical application, as it may efficiently decrease MAPK-ERK activation caused by IL-7R/JAK mutations or physiological IL-7 transmission, whereas it is not useful in the alterations occurring downstream of IL-7R/JAK. Such limitations are not encountered when using MAPK inhibitors [118].

Mutations of genes encoding the rat sarcoma virus (Ras) protein family are found in around 30% of all cancers in humans, making them the most common genetical alterations occurring in cancer [117,119]. Ras’s proteins are involved in the Ras/rapidly accelerated fibrosarcoma (Raf)/MAPK/ERK signaling cascade; thus, Ras activation leads to ERK induction [119]. Irving et al. examined 54 samples obtained from children with pre-B ALL at the time of diagnosis in search for activating mutations of Ras proteins (*KRAS* and *NRAS*), mutations of Ras regulators (*PTPN11),* and alterations in genes encoding upstream signaling proteins *(FLT3).* In 28 (51.9%) children, such mutations have been found. Ras/Raf/MAPK/ERK pathway activation was estimated in 80 pre-B ALL patients. Among 32 cases with Ras-related mutations, 27 (84.3%) had relevant Ras/Raf/MAPK/ERK signaling cascade activation. In the remaining patients without Ras mutations, activation of this pathway was found in 9 of 48 (18.7%) samples. The cells with Ras/Raf/MAPK/ERK activation were significantly more sensitive to selumetinib, regardless of the presence of Ras-related mutations [120]. Jerchel et al. found Ras-related mutations (*NRAS*, *KRAS*, *FLT3*, *PTPN11,* and others) in 44.2% of 461 samples obtained from children with pre-B ALL at the time of diagnosis. Overall, the Ras-related mutations were most frequent in the high hyperdiploid (72.6%) and t (4;11)-rearranged (73.3%) pre-B ALL. Ras-related mutations were associated with significantly worse clinical outcomes in the HR patients. It was further established that Ras mutations are associated with glucocorticoid resistance, as samples with Ras-related mutations were 3-fold more resistant to prednisolone treatment than samples without these mutations (*p* = 0.024). In this context, the clonal/subclonal *KRAS* G13 were most harmful, as blasts with this mutation were characterized by more than 2000-fold-higher glucocorticoid resistance, as compared to wild-type leukemic cells. In contrast, *NRAS* and *KRAS* G12 were found not to increase glucocorticoid resistance in a significant manner. The treatment of Ras-mutated samples using MAPK inhibitor trametinib induced cytotoxicity, whereas wild-type blasts were unaffected (*p* = 0.001) [121]. Signaling cascades involved in the glucocorticoid resistance in ALL have been schematically presented on Figure 1.

Matheson et al. demonstrated, that selemutinib and dexamethasone co-treatment is highly synergic, both in the in vitro and in vivo studies. In vitro, selemutinib and dexamethasone co-treatment exerted a highly synergic impact (mean combination index (CI) of 0.1) on primagrafts with Ras-related mutations (affecting *NRAS*, *KRAS*, and *CBL/FLT3*). Combined therapy resulted in elevation of BIM levels, a decreased Mcl-1 level, and ERK phosphorylation. In vivo, it was demonstrated that selumetinib and dexamethasone co-treatment acts highly synergic in nonobese diabetic (NOD) SCID (severe combined immunodeficiency) gamma mice engrafted with a Ras-mutated, patient-derived xenograft. *KRAS G13D*, *KRAS* G12D, and *NRAS* Q61R mice treated with drug combination had a spleen weight at the end of the therapy comparable with the healthy mice spleen (*p* < 0.001) [122]. Polak et al. obtained blast samples from 22 adults with newly diagnosed pre-B ALL. These cells were incubated with dexamethasone or dexamethasone in combination with selumetinib. Combined therapy managed to augment the dexamethasone-induced apoptosis in 17 samples [123].

*Ras*-activating mutations have been also found in the relapsed T-ALL patients and were associated with highly unfavorable outcomes [89,124,125,126,127]. Kerstjens et al. treated *Ras* mutant and *Ras* wild-type t (4;11)^+^ infant ALL blasts with salirasib (Ras inhibitor), vemurafenib (serine/threonine-protein kinase B-raf inhibitor), sorafenib (pan-kinase inhibitor), temsirolimus (mTOR inhibitor), and MAPK inhibitors trametinib, selumetinib and binimetinib. MAPK inhibitors were significantly more efficient than the other tested drugs [128]. This is particularly important, as *Ras*-mutated mixed lineage leukemia (MLL)-rearranged leukemic cells are likely to be resistant to glucocorticoid treatment [129]. MLL-rearranged ALL is also associated with far worse clinical outcomes as compared to patients without MLL translocations [130].

Combining JAK inhibitors with blockers of the downstream signaling also seems to be a promising strategy for the ALL treatment. Ba/F3 cells harboring JAK3(L857Q) and JAK3(M511I) mutations were treated with tofacitinib (JAK1/JAK3 inhibitor) and selumetinib at various concentrations. All combinations were characterized by synergic, dose-dependent action (CI < 0.1–0.9). A tofacitinib and venetoclax combination was also synergistic, however to a lesser extent. Co-treatment with tofacitinib and buparlisib (PI3K inhibitor) also provided mediocre effects. Furthermore, in the blasts harboring a *JAK3 (M511I)* mutation, obtained from the T-ALL patient, the combination of tofacitinib and trametinib was highly synergic (CI from <0.1 to 1, depending on the drugs concentrations). Further, tofacitinib and venetoclax co-treatment at low concentrations was also effective (CI < 0.1). Oral treatment consisting of venetoclax (20 mg/kg/day) and ruxolitinib (40 mg/kg/day) was effective in treating mice engrafted with blasts carrying the *JAK3 (M511I)* mutation [131]. Inhibiting the JAK/STAT pathway, combined with blocking MAPK/ERK signaling, seems to be a rational strategy. MAPK/ERK can be also activated by *Ras*-activating mutations, therefore blocking JAK/STAT signaling may not be enough to overcome resistance to the treatment [118,119,120,121,122,123]. Furthermore, combining JAK/STAT inhibitors with Bcl-2 inhibitors is also rational, as STAT proteins induce Bcl-2. Unfortunately, MAPK inhibitors have not been used in co-treatment with venetoclax in this study, as this combination could also be beneficial. Further, using all of these drugs simultaneously could hypothetically provide the best results, as it would target various mechanisms of treatment resistance. Thus, further preclinical studies are required.

Agents capable of blocking signaling pathways involved in the ALL glucocorticoid resistance have been systematized in the Table 1.

## 4. Methods of Enhancing the Results of Glucocorticoid Therapy in Pediatric ALL

### 4.1. Enhancing Effects of GR Activation

Roderick et al. showed that *NR3C1* mRNA and GR protein levels are increased by cyclic adenosine monophosphate-dependent protein kinase (cAMP-PKA) signaling in mouse and human T-ALL cells. Furthermore, when the cAMP-activating *Gnas* gene was silenced, it caused resistance to the dexamethasone treatment, both in vitro and in vivo. Further, cAMP activation induced by 6,16-dimethyl-prostaglandin E2 (dmPGE2) and dexamethasone co-treatment was found to overcome glucocorticoid resistance in the T-ALL patient samples. Therefore, it was concluded that the cAMP-PKA-induced increase in GR level is the reason for its beneficial effects in co-treatment with dexamethasone [134]. This interesting and novel mechanism requires further laboratory research.

The microRNAs (miRNAs) miR-100 and miR-99a were shown to be downregulated in childhood ALL patients, and their expression levels were linked to the ALL patients’ prognosis, as demonstrated by the investigation by Li et al. MiR-100 and miR-99a were found to be important in the regulation of cell proliferation and dexamethasone-induced apoptosis in ALL cell lines in vitro. The researchers discovered that the FK506-binding protein 51 (FKBP51) is a novel target of miR-100 and miR-99a. FKBP51, in turn, is responsible for inhibiting GR’s nuclear translocation. The findings of the study demonstrate that miR-100 and miR-99a are tumor suppressors, and that reinstating them could be a treatment option for ALL patients [135]. A study by Liang et al. showed that miR-124 expression is significantly increased in children suffering from glucocorticoid-resistant ALL, as compared to the glucocorticoid-sensitive samples. Moreover, miR-124 causes dexamethasone resistance and suppresses glucocorticoid-induced apoptosis in the sensitive cell lines CCRF-CEM and CEM/C1. miR-124 targets NR3C1, suppressing its expression, consequently lowering efficiency of glucocorticoid treatment. Thus, Liang et al. proposed a unique mechanism for GC resistance in ALL, which could be potentially targeted in order to overcome treatment resistance [136].

Paugh et al. discovered that the recombinant caspase 1 (CASP1) cleaves the GR at its transactivation region, and that forced overexpression of CASP1 combined with the NLR family pyrin domain containing three inflammasome activations causes human leukemia cells to become resistant to glucocorticoid therapy. Persistently inhibiting CASP1 expression or lowering CASP1 activity with an inhibitory protein (CrmA) in CASP1-overexpressing leukemia cells raises cellular GR levels and significantly enhances glucocorticoid sensitivity [137]. Therefore, CASP1 inhibitors (e.g., VX 765 and Ac-FLTD-CMK), or even pan-caspase inhibitors such as emricasan, which is clinically used in the treatment of non-alcoholic fatty liver disease, could potentially be implemented in the ALL therapy.

Gallagher et al. discovered that the orphan nuclear receptor estrogen-related receptor-b (ESRRB), is an important transcription factor that collaborates with the GR to mediate the GTGs expression signature in mice and human ALL cells using a genome-wide, survival-based short hairpin RNA (shRNA) screen. They found that ESRRB knockdown inhibits dexamethasone-induced gene expression, implying that ESRRB cooperates with GR to generate optimal dexamethasone transcriptional responses. Furthermore, ESRRB agonist (GSK4716), in combination with dexamethasone, is highly synergic—a CI of <1 in treating human T-ALL cell lines KOPTK1 and DND-41 [138].

### 4.2. The BH3 Mimetics—Targeting the Primary Mechanism of Glucocorticoid-Induced Apoptosis of ALL Cells

The BH3 mimetics, which are the drugs that initiate apoptosis by targeting pro-survival Bcl-2 proteins, show great potential in the treatment of ALL. Venetoclax (previously ABT-199), a discovered 2013 Bcl-2 inhibitor, is currently the only BH3 mimetic that has been clinically approved [139,140]. Both Food and Drug Administration (FDA) and European Medicines Agency (EMA)-approved venetoclax for the treatment of chronic lymphocytic leukemia and acute myeloid leukemia (AML) [140,141,142]. Navitoclax (ABT-263) is a BH3 mimetic which inhibits not only Bcl-2, but also Bcl-XL and Bcl-w. Unfortunately, navitoclax has not been approved yet by the FDA and EMA for clinical use. However, navitoclax in co-treatment with venetoclax is available for expanded access (NCT03592576). Therefore, there are prospects of navitoclax approbation for co-treatment with different BH3 agonists.

A particularly interesting study has been conducted by Ni Chonghaile et al., who demonstrated that leukemic cells in different maturation stages have dissimilar levels of Bcl-2 proteins. Cell lines underwent Bcl-2 profiling, which was based on the strong Bcl-2 binding to Bad (but not to Hrk), whereas Bcl-XL interact equally with Bad and Hrk. Most T-ALL cell lines (CEM-CCRF, PF382, Molt4, P12-Itchikawa, Jurkat, KOPTKI, RPMI-1640, and CEM-C1) were characterized by a strong Bcl-XL dependence, which means that Bcl-XL was mainly responsible for anti-apoptotic action and, consequently, cells’ survival. In contrast, ETP, the Loucy cell line, was found to be Bcl-2 dependent. Venetoclax sufficiently induced apoptosis in Loucy cells, however, navitoclax was also effective. Unsurprisingly, Bcl-XL-dependent cells were killed more efficiently by navitoclax. Furthermore, 26 samples obtained from primary pediatric T-ALL (10 with ETP-ALL) at the time of diagnosis were examined. The ETP-ALL cells’ survival was definitely Bcl-2 dependent, in contrast with typical T-ALL, which was found to be Bcl-XL dependent, thus confirming the results obtained in vitro. Moreover, healthy cells at the earliest double negative (DN) intrathymic stage of differentiation (DN1) were also found to be Bcl-2 dependent, corresponding to ETP-ALL leukemic cells. Cells during the CD4^+^ and CD8^+^ stages of differentiation were, in turn, dependent of Bcl-XL, thus indicating that there is a significant change in expression of Bcl-2 family proteins during the lymphocyte development. A Bcl-2 profiling of the T-ALL samples obtained during the Dana-Farber Cancer Institute trials additionally confirmed that ETP-ALL cells are dependent on Bcl-2, whereas typical T-ALL was characterized by the Bcl-XL dependence. Similarly, to the previous outcomes, in this case, ETP-ALL was also sensitive both to venetoclax and navitoclax, whereas primary T-ALL cells responded well to navitoclax. Patient-derived xenografts from mature T-ALL and ETP-ALL patients were established, and mice with at least 65% CD45+ blasts were given venetoclax, navitoclax, or a vehicle for 14 days. The results were comparable to previous tests, as ETP-ALL xenografts were sensitive to both venetoclax and navitoclax, whereas typical T-ALL xenografts responded better to navitoclax than venetoclax [143]. A study conducted by Peirs et al. evaluated in vitro Bcl-2 expression in different T-ALL molecular subgroups, as well as their response to venetoclax treatment. Mature T-ALL cell lines ALL-SIL, CUTLL1, TALL-1, KOPTK1, DND-41, PF-382, KARPAS-45, PEER, Jurkat, and CCRF-CEM were characterized by intermediate responses to venetoclax treatment, whereas Loucy cells were highly sensitive to venetoclax, which confirms the results obtained by Ni Chonghaile et al. Seventeen pediatric primary T-ALL bone marrow samples, with different maturation states and genetical profiles, were selected, and the cells were treated with venetoclax. Cells in the earlier stages of maturation were characterized by half maximal inhibitory concentration (IC_50_) < 300 nM, whereas in most of the mature cells, the treatment outcome was worse [144]. Venetoclax is currently being tested in combination with C10403 chemotherapy for the newly diagnosed adult pre-B ALL during the phase I study (NCT05157971). The future prospects of BH3 mimetics used in ALL have been summarized in Table 2.

### 4.3. Proteasome Inhibitors—Multitude Mechanism of Action Benefiting the Glucocorticoid Treatment

Proteasome inhibitors showed great potential in treating cancer in multiple studies. Currently, three drugs belonging to this group have been approved for clinical use by the FDA and EMA: bortezomib, ixazomib, and carfilzomib [148]. Bortezomib is an efficient reversible inhibitor of the β5 chymotrypsin-like subunit of 20S proteolytic site of the 26S proteasome. Bortezomib also reversibly blocks the β1 caspase-like subunit and β2 trypsin-like subunit when used in higher doses [148,149]. Inhibition of proteasomes results in higher p27 and p53 levels. Additionally, proteasomes are crucial in activation of nuclear factor-κB (NF-κB), which ceases transcription of pro-apoptotic factors [150]. Moreover, bortezomib activates c-Jun N-terminal kinase and cause aggregation of incorrect proteins, which also promotes apoptosis [151]. Results of the study on the effects of prednisolone and bortezomib co-treatment of pre-B leukemic cells have been published. Both glucocorticoid-resistant (MHH-cALL-2) and sensitive (MHH-cALL-3) cells underwent treatment with single agents or co-treatment with both drugs. Bortezomib alone induced apoptosis in both cell lines in a dose-dependent manner. In both cell lines, bortezomib and prednisolone co-treatment caused additional increase in cell death, mostly in MHH-cALL-2, indicating that bortezomib may be useful in overcoming glucocorticoid resistance [152]. It should be noted that bortezomib may also increase Noxa levels, contributing to enhancing apoptosis in glucocorticoid-resistant cells via decreasing glucocorticoid-induced proteasomal degradation of Noxa [62]. This hypothesis is supported by results of several in vitro studies, in which bortezomib was found to increase Noxa levels [153,154,155,156]. Nevertheless, this potential effect of bortezomib and other proteasome inhibitors’ administration in ALL cells requires further research.

Carfilzomib is a drug belonging to a second generation of proteasome inhibitors, which is an irreversible blocker of β5/β5i subunits [149]. Carfilzomib in a single treatment exhibited an antileukemic action in the Molt4 cells (a glucocorticoid-resistant human T-ALL-derived cell line). It has also sensitized the Molt4 cell line to dexamethasone-induced apoptosis. Interestingly, carfilzomib caused the generation of reactive oxygen species (ROS), thus generating the state of oxidative stress in the Molt4 cells [157]. ROS generation caused forkhead box O3 (FOXO3a) induction, which causes cell death via up-regulation of autophagy-related genes [157,158]. Carfilzomib-generated ROS production also elevates the stress induced sirtuin 1 (SIRT1) level, which plays a role in the endoplasmic reticulum’s (RER) stress-induced apoptosis and autophagy [157,159]. Another interesting mechanism of the carfilzomib action is increasing the C/EBP Homologous Protein (CHOP) protein transcription, which is a downstream target of protein kinase R (PKR)-like endoplasmic reticulum kinase (PERK) and the cyclic AMP-dependent transcription factor (ATF4) [157]. CHOP acts as a critical pro-apoptotic mediator in the RER stress, and when activated, it regulates the Bcl-2 family proteins. In the case of RER stress, CHOP decreases Bcl-2, Bcl-XL, and Mcl-1 levels, whereas it raises the *BIM* expression. Additionally, CHOP induce tribbles homolog 3 (TRB3), which in turn inhibits AKT activity. CHOP has been also demonstrated to up-regulate expression of death receptor (DR) 4 and DR5, which are the components of the DR pathway [160]. CHOP was also found to activate GADD34, a DNA damage protein. CHOP may also be activated in the course of bortezomib-induced RER stress in different cancer cell lines [161,162,163].

Unfortunately, for carfilzomib combined with dexamethasone, mitoxantrone, methotrexate, pegylated L-asparaginase, and vincristine (UKALLR3), induction therapy was found to be excessively toxic [164]. Interestingly, phase I of the carfilzomib and hyperfractionated cyclophosphamide, vincristine, doxorubicin, and dexamethasone (HyperCVAD) co-treatment in newly diagnosed ALL has been recently completed. Ten patients with Philadelphia negative ALL (eight with pre-B ALL and two with T-ALL), aged 18–64, were enrolled. The treatment was well-tolerated and resulted in a complete response (CR) rate of 90% after the first cycle; the last patient achieved CR after the fourth cycle. Moreover, MRD negativity was achieved in seven (70%) patients [165].

Ixazomib is a drug belonging to the third generation of proteasome inhibitors, which acts mainly by reversibly blocking β5/β5i subunits; thus, its action is similar to bortezomib. However, it lasts significantly shorter. Ixazomib also reversibly inhibits β1 and β2 subunits [148,149]. This drug has been found to inhibit growth of the wild-type CCRF-CEM T-ALL; however, bortezomib was found to be 10 times more potent. Furthermore, the leukemic cells obtained from nine patients suffering from primary ALL were significantly more sensitive to bortezomib as compared to ixazomib (*p* < 0.0001) [166]. Nevertheless, it was suggested that this difference in effective concentration of these drugs during in vitro studies may not be present in the in vivo studies. Ixazomib is characterized by excellent pharmacokinetics properties; it is quickly absorbed from the digestive tract and its tissue distribution is 5-fold better than bortezomib’s [148,166]. Therefore, the in vivo results of ixazomib treatment may be more promising [166].

The high potential of bortezomib and carfilzomib in the ALL treatment is reflected by the considerable number of currently ongoing clinical trials (Table 3).

## 5. Other Prospects of Enhancing Glucocorticoid Sensitivity in ALL

### 5.1. Tigecycline

Tigecycline is the first in a new class of antibiotics, the glycylcyclines, which are structurally related to tetracyclines. Tigecycline inhibits the protein translation in bacterial cells by binding the 30S ribosome subunit and blocking the attachment of the aminoacyl-tRNA molecule to the ribosomal A-site [172]. It is highly active against many gram-positive and gram-negative pathogens, both aerobic and anaerobic. As bacterial and mitochondrial ribosomes are similar, tigecycline also inhibits the synthesis of mitochondria-encoded proteins involved in oxidative phosphorylation (OXPHOS). It has been reported that such a mechanism of action of tigecycline is responsible for its properties of inhibiting development of cancers, such as chronic myeloid leukemia stem cells, ALL, non-small cell lung cancer, ovarian cancer, and hepatocellular carcinoma [173,174]. Enhanced oxidative phosphorylation is associated with steroid resistance in ALL cells. Studies revealed that inhibitors of OXPHOS pathways increase sensitivity to steroids in T-ALL in vitro [175]. Therefore, research on the use of tigecycline in ALL therapeutic protocols are suggested. So far, Xuedong et al. reported preclinical evidence that tigecycline suppresses mitochondrial respiration, induces apoptosis of cancer cells, and also intensifies the effect of chemotherapeutic drugs—doxorubicin and vincristine [176]. These conclusions may be a starting point for further studies.

### 5.2. Tamoxifen

Tamoxifen is a synthetic anti-estrogenic medicine used mainly in the treatment of luminal breast cancer. Recently, new targets of tamoxifen activity other than estrogen receptors have been discovered. In the literature, attention is paid to its influence on important mediators of signaling pathways initiating cell proliferation, conditioning the aggressive course of disease, or cancer sensitivity to chemotherapy [177]. Taking these findings into account, the possibility of using tamoxifen in the therapy of neoplasms other than breast cancer should be considered.

As it was mentioned previously (Section 3.1), resistance to steroids correlates with increased activity of the anti-apoptotic proteins of the Bcl-2 family. Tamoxifen has been demonstrated to stimulate apoptosis by reducing the expression of Bcl-2, as well as increasing the expression of the pro-apoptotic Bax protein [177]. Another reported effect of tamoxifen is the activation of autophagy, confirmed by experiments conducted on various tumor cell lines [178]. Autophagy is a natural process in which defective or dispensable intracellular proteins or cell organelles are delivered to lysosomes for degradation. In cancer cells, autophagy plays a double role—it may be responsible for the development of resistance to the applied therapy, as well as for cell death. On the one hand, it can be activated in response to stress factors such as radiation therapy or chemotherapy and lead to tumor growth and survival. Autophagy can inhibit neoplastic transformation and induce apoptosis, and consequently increase the effectiveness of the treatment [179]. Since glucocorticoids induces cell death through the autophagic machinery activation, tamoxifen, which enhances this process, may be an effective medicine in partial reversing steroid resistance [180].

### 5.3. Cannabidiol

For many years, cannabidiol (CBD) has been used in the treatment of drug-resistant epilepsy [181]. The FDA has approved it for the treatment of Lennox–Gaustat and Dravet syndromes. Compared to Δ9-tetrahydrocannabinol (Δ9-THC), CBD has a weak affinity for cannabinoid receptor type 1 and type 2 receptors, and, therefore, does not cause undesirable side effects on the central nervous system (CNS). In oncology, cannabidiol is an aid in palliative treatment. It alleviates the symptoms associated with cancer and chemotherapy, such as pain, nausea, vomiting, loss of appetite, or even anxiety. Recently, however, cytotoxic effects of cannabinoids have been discussed. They have been shown to have pro-apoptotic and antiproliferative effects [182]. Rimmerman et al., in their study, suggest that cannabinoids may cause cancer cell death by modulating the voltage-dependent anion channel 1 (VDAC) located in the outer mitochondrial membrane [183]. VDAC enables the transfer of ions and metabolites between the cytoplasm and the mitochondria, thus regulating many processes, such as apoptosis and cell metabolism. Besides, it anchors many proteins, including hexokinase. This combination is responsible for the regulation of glycolysis and protection against cell death [184]. CBD binds to mitochondrial membranes containing VDAC, thereby reducing the conductivity of the channels, weakening cell viability and ultimately leading to apoptosis [183]. VDAC plays a key role in the coordination between the aforementioned oxygen glycolysis and mitochondrial OXPHOS. It is worth considering the use of a combination of glucocorticoids with CBD in anti-cancer therapy, which could increase sensitivity to steroid therapy [185].

### 5.4. Mebendazole

Anthelmintics are given to children suffering from ALL, as they are more prone to parasitic infections while receiving chemotherapy. However, their use may also be justified by their anti-cancer activity. Mebendazole has been shown to inhibit the neurogenic locus notch homolog protein (Notch) signalization consisting of four receptors (Notch 1–4) [186]. Arresting Notch signaling can reverse glucocorticoid resistance in T-ALL by restoring autoregulation of the GR and induction of the gene-encoding Bcl-2 [187].

### 5.5. Demethylating Agents

*BIM* IGR exhibits inactive chromatin configuration in non-lymphoid cells, whereas the lymphoblasts have accessible chromatin at this site, making *BIM* IGR a lymphocyte-specific target of GR activation. Moreover, glucocorticoid-resistant lymphoblasts are characterized by the high *BIM* IGR methylation, and, consequently, low chromatin accessibility impairing GR’s binding, in contrary to glucocorticoid-sensitive cells, which have a low level of *BIM* IGR methylation and are well accessible. Furthermore, treatment of ALL-7CL-resistant cells with demethylating drug 5-azacitidine resulted in a time-dependent decrease in *BIM* IGR methylation. Moreover, co-treatment of ALL-7CL with dexamethasone and 5-azacitidine caused a significant increase in *BIM* expression after 48 h, compared to ALL-7CL treated only with dexamethasone (however, it was pretreated with 5-azacitidine for 3 days prior to dexamethasone). When ALL-7R-engrafted, NOD SCID gamma mice underwent 14 days of treatment with dexamethasone combined with decitabine, a clinically used demethylating agent, there was a significant reduction in leukemic cells in bone marrow combined with longer EFS, as compared to the mice in which only one agent was used. This indicates the critical importance of opposing DNA methylation in alleviating glucocorticoid resistance in ALL [188].

## 6. Conclusions and Future Directions

In conclusion, most of the preclinical studies involve novel agents that are capable of enhancing the results of glucocorticoid therapy. It should be emphasized that the described drugs could potentially be included in the induction phase in the newly diagnosed ALL patients. This, ultimately, could prevent relapse and provide better outcomes. Furthermore, incorporating into the diagnostics BH3 profiling and screening for activating mutations in the IL7/JAK/STAT cascade, the Ras pathway, downstream PI3K/AKT/mTOR, and MAPK/ERK pathways could potentially help to identify the patients that could benefit from targeting those signaling cascades. Nevertheless, the described findings still require further research.

## Figures and Tables

**Figure 1 ijms-23-03795-f001:**
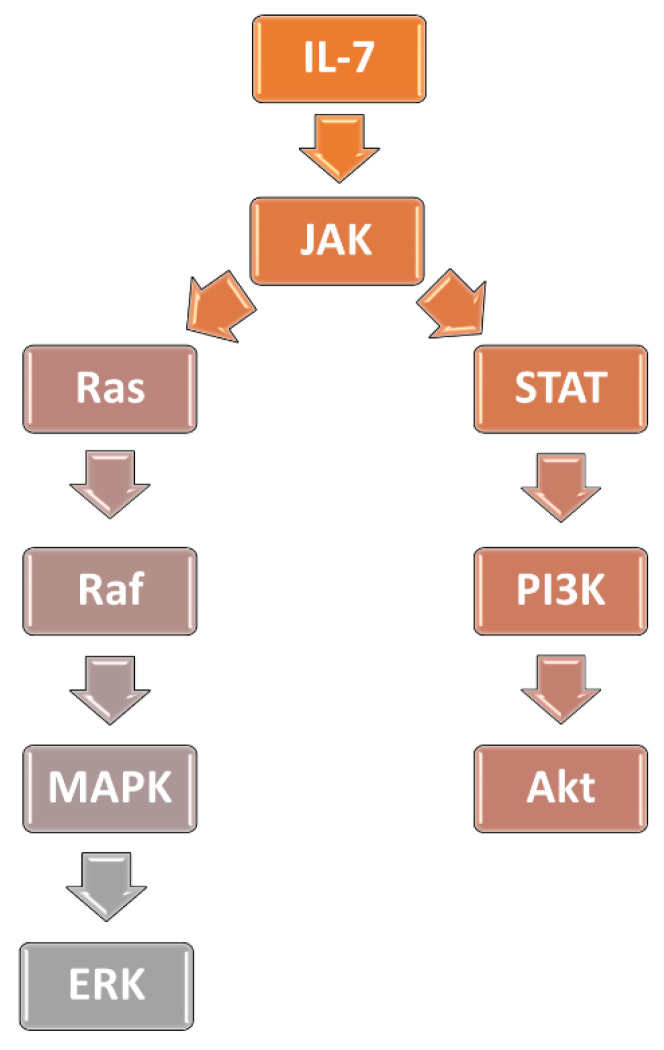
Signaling cascades involved in glucocorticoid resistance.

**Table 1 ijms-23-03795-t001:** Targeting signaling pathways in ALL treatment.

Drugs	Mechanism of Action	Preclinical Studies	Completed Clinical Studies	Ongoing Clinical Studies
MK2206	Allosteric AKT ^1^ inhibition	MK2206 and dexamethasone co-treatment of resistant ALL ^2^ cell lines in vitro and in vivo [99]	Not applicable	Not applicable
Pictilisib Idelalisib Buparlisib ZSTK-474 AS-605240 CAL-101 Duvelisib	pan-PI3K ^3^ inhibition	Pictilisib in T-ALL ^4^ treatment [102,103,104,105] Idelalisib treatment of T-cell leukemia-lymphoma samples [107]	Not applicable	Phase I, chimeric antigen receptor T-cell followed by duvelisib for ALL, LL ^5^, and lymphosarcoma (NCT05044039). Phase I, duvelisib for relapsing/remitting T-LL ^7^
		Buparlisib treatment of T-ALL cell lines [108,109] ZSTK-474, AS-605240, CAL-101, and IPI-145 for the treatment of pre-B ALL ^8^ cell lines and ex vivo blasts from pre-B ALL patients [110]		
Dactolisib	pan-PI3K inhibition, mTOR ^9^ inhibition	Dactolisib treatment of T-ALL and pre-B ALL cell lines [132]	Phase I, relapsed/remitting ALL [115]	Not applicable
Ruxolitinib	JAK1/2 inhibition	Ruxolitinib and dexamethasone co-treatment of T-ALL cells [90] Ruxolitinib and VXL ^10^ chemotherapy in mice engrafted with Ph-like ^11^ ALL [96] Ruxolitinib treatment of cell lines with IL-7 ^12^-signaling, mediated steroid resistance. Ruxolitinib alone or in co-treatment with prednisolone in the T-ALL xenografts ex vivo [118] Venetoclax and ruxolitinib co-treatment in vivo in treating mice engrafted with blasts carrying JAK3 mutation [131]	Phase II—part 1, ruxolitinib and consolidation chemotherapy for pediatric Ph-like ALL [133]	Phase I, newly diagnosed pediatric Ph-like ALL (NCT03571321) Phase II—part 2, pediatric Ph-like ALL (NCT02723994) Phase II, relapsing/remitting Ph-like ALL (NCT02420717) Phase II/III, pediatric T-ALL/T-LL or pre-B ALL/B-LL ^13^ (NCT03117751)
CI1040	MAPK ^14^ inhibition	CI1040 treatment of glucocorticoid-resistant cell lines [89] CI1040 treatment of cell lines with IL-7-signaling, mediated steroid resistance [118]	Not applicable	Not applicable
Selumetinib Trametinib Binimetinib	MAPK inhibition	Selumetinib and trametinib treatment of cell lines with IL-7-signaling, mediated steroid resistance. Selumetinib, trametinib and binimetinib alone in the T-ALL xenografts ex vivo. Selumetinib in co-treatment with prednisolone alone in the T-ALL xenografts ex vivo [118] Selumetinib ex vivo treatment of pre-B ALL samples. Selumetinib in vivo treatment of xenografts with Ras ^15^ pathway mutant/wild-type ALL cells [120] Selumetinib in vivo and in vitro treatment of xenografts with Ras pathway mutant ALL cells [122] Selumetinib in vivo and in vitro treatment of xenografts with Ras pathway mutant/wild-type ALL cells [122] Selumetinib in co-treatment with dexamethasone for pre-B ALL samples ex vivo and in vitro pre-B ALL and T-ALL cell lines [123] Trametinib treatment ex vivo for pre-B ALL samples [121] Selumetinib, trametinib and binimetinib treatment of Ras mutated/wild-type MLL ^16^ rearranged ALL cell lines [128] Tofacitinib and MAPK inhibitors co-treatment for JAK ^6^-mutated ALL cell lines and ex vivo blasts from T-ALL patients [131]	Not applicable	Phase I/II, selumetinib for the relapsing/remitting ALL (NCT03705507)

^1^ AKT, serine/threonine kinase; ^2^ ALL, acute lymphoblastic leukemia; ^3^ PI3K, phosphatidylinositol 3-kinase; ^4^ T-ALL, precursor T-cell acute lymphoblastic leukemia; ^5^ LL, lymphoblastic lymphoma; ^6^ JAK, Janus kinases; ^7^ T-LL, T-lymphoblastic lymphoma; ^8^ pre-B ALL, precursor-B-cell acute lymphoblastic leukemia; ^9^ mTOR, mammalian target of rapamycin kinase; ^10^ VXL, vincristine, dexamethasone and L-asparaginase; ^11^ Ph-like, Philadelphia chromosome-like; ^12^ IL-7, interleukin-7; ^13^ B-LL, B-lymphoblastic lymphoma; ^14^ MAPK, mitogen-activated protein kinase; ^15^ Ras, rat sarcoma virus; ^16^ MLL, mixed lineage leukemia.

**Table 2 ijms-23-03795-t002:** BH3 mimetics in ALL treatment.

Drug	Preclinical Studies	Completed Clinical Studies	Ongoing Clinical Studies
Venetoclax	Venetoclax treatment of T-ALL ^1^ and ETP ALL ^2^ cell lines/blasts from patients [143] Venetoclax treatment of T-ALL cell lines/blasts from patients [144]	Series of cases, relapsing/remitting ALL ^3^ [145] Retrospective study, relapsing/remitting T-ALL [146] Phase I, relapsing/remitting ALL [147]	Phase Ib-II, navitoclax and venetoclax co-treatment for pre-transplant and post-transplant treatment of adult T-ALL patients (NCT05054465) Phase I, adult pre-B ALL ^4^ (NCT05157971) Phase I/phase II, relapsing/remitting ALL (NCT03808610, NCT03504644, NCT03576547, NCT03319901, NCT04872790, NCT05016947, NCT03808610, NCT04752163, and NCT05149378)
	Venetoclax and tofacitinib ex vivo co-treatment of JAK ^5^-mutated ALL cell lines and blasts from T-ALL patients [137]		
Navitoclax	Navitoclax treatment of T-ALL and ETP ALL cell lines/blasts from patients [143]	Phase I, relapsing/remitting ALL [147]	Phase Ib-II, navitoclax and venetoclax co-treatment for pre-transplant and post-transplant treatment of adult T-ALL patients (NCT05054465)

^1^ T-ALL, precursor T-cell acute lymphoblastic leukemia; ^2^ ETP ALL, early T-cell progenitor acute lymphoblastic leukemia; ^3^ ALL, acute lymphoblastic leukemia; ^4^ pre-B ALL, precursor-B-cell acute lymphoblastic leukemia; ^5^ JAK, Janus kinases.

**Table 3 ijms-23-03795-t003:** Proteasome inhibitors in ALL treatment.

**Drug**	**Preclinical Studies**	**Completed Clinical Studies**	**Ongoing Clinical Studies**
Bortezomib	Bortezomib and prednisone co-treatment of resistant/sensitive ALL ^1^ cell lines [152] Bortezomib treatment of sensitive T-ALL cell line and cells from primary ALL patients [166]	Phase I/II, pediatric relapsed ALL [167,168] Prospective cohort study, pediatric relapsed ALL [169] Phase II, relapsed ALL/T-LL [170]	Phase II, infants with newly diagnosed ALL (NCT02553460) Phase II/III, pediatric T-ALL ^2^/T-LL ^3^ or pre-B ALL ^4^/B-LL ^5^ (NCT03117751) Phase III, newly diagnosed ALL/LL ^6^ (NCT02112916) AIEOP-BFM ALL 2017 study, early- HR ^7^ pre-B ALL (NCT03390387) Phase II, relapsed/remitting ALL (NCT03136146, NCT03590171) Phase IV, relapsed ALL (NCT05137860)
Carfilzomib	Carfilzomib and dexamethasone co-treatment in resistant cell line [157]	Phase I, relapsed/refractory ALL [171]	Phase II, relapsed/refractory ALL (NCT02303821)
		Phase I, newly diagnosed ALL [165]	Phase I, relapsed/refractory solid tumors or leukemia (NCT02512926)
Ixazomib	Ixazomib treatment of sensitive T-ALL cell line and cells from primary ALL patients [166]	Not applicable	Phase I/II, relapsed/remitting ALL (NCT03817320)

^1^ ALL, acute lymphoblastic leukemia; ^2^ T-ALL, precursor T-cell acute lymphoblastic leukemia; ^3^ T-LL, T-lymphoblastic lymphoma; ^4^ pre-B ALL, precursor-B-cell acute lymphoblastic leukemia; ^5^ B-LL, B-lymphoblastic lymphoma; ^6^ LL lymphoblastic lymphoma; ^7^ HR, high risk.

## Data Availability

No new data were created or analyzed in this study. Data sharing is not applicable to this article.

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
