# Peer review of "Overcoming Steroid Resistance in Pediatric Acute Lymphoblastic Leukemia—The State-of-the-Art Knowledge and Future Prospects"

_ijms, 2022, doi:10.3390/ijms23073795_

Round 1

Reviewer 1 Report

Kosmider K et al submitted an extended review on "Overcoming steroid resistance in Ped ALL".

Major comment

The treated arguments are misleading, although appropriate for ALL. It seems to be a review on "new treatment for ALL", starting from molecular and cellular pathways. The authors wrote deeply and too extensively on biological aspects, which should be the case based on the journal's policy, citing many protocols and different strategies which pertain to treatment of ALL.

Therefore, my suggestion is to completely revise the manuscript, trying to re-write the article throughout the main argument "glucocorticoid resistance" on pediatric ALL without citing any other form of new treatment. It is undoubted that glucocorticoid resistance is tightly related to relapse, but if the main goal of this review is "stored resistance" the authors have to address this specific item, not extending and stretching too much the concepts of resistance. Readers will be "lost in translation".

Please shorten the entire manuscript, starting from Introduction. Tables should be cited in the text earlier than the "conclusion" section.

I would also suggest to consider a re-organization of the manuscript. Starting from the mechanism of action, the authors should show the latest results only regarding new treatment for overcoming steroid resistance and not relapses or recurrent disease. I invite the authors to focus specifically on biological mechanisms, translated to clinical application but only for steroid-resistance. The article will be shortened and the "story-telling" will be more fluid and well comprehensible.

After that effort, I will be glad to review it again.  

Reviewer 2 Report

In the review entitled "Overcoming steroid resistance in pediatric acute lymphoblastic leukemia – the state-of-the-art knowledge and future prospects" by KoÅ›mider Kamil and colleagues, authors give comprehensive overview of the current knowledge of corticosteroid resistance in pediatric ALL. The manuscript is detailed in describing mechanisms of steroid resistance and current challenges in overcoming it. In that respect, the manuscript could benefit from being more selective when describing the details of each study. Some of these details do not contribute to the overall quality of the manuscript, while at the same time greatly increase its size, which can be off-putting to the reader. Other than this, the manuscript gives solid basis for readers interested in the mechanisms of steroid resistance. 

Major comments:

1) Authors should avoid using hard and definite terms such as "proven" and use instead softer terms such as "shown" or "demonstrated" when referring to individual studies. Saying that something is proven would mean that this is definite and no further conclusions or alternate findings are possible, which is rarely the case in science. 

2) In line with previous comment, in the lines 1023-1026 authors say that it was proven that in a significant group of children minor subclones in RAS pathway genes may contribute to relapse development. In fact, current studies do not offer many evidence on subclonal mutations causing relapse in ALL, but rather give contradictory findings. For example, in the study from Jerchel et al, which is discussed in the next paragraph of the manuscript, authors have shown that subclonal KRAS mutations do not contribute to relapse development. In another Dutch study, from Antić et al (PMID: 33147938), authors assessed clinical relevance of subclonal mutations in a group of relapse associated genes and did not find association with relapse or unfavorable outcome in patients which had subclonal RAS mutations. My opinion is that this part needs to be written more carefully and findings from other studies need to be discussed.

3) Authors should discuss deletions of IKZF1 gene in the context of steroid resistance. Bearing in mind that IKZF1 deletions/IKZF1plus status is used as an important risk stratification marker in the contemporary treatment protocols, as well as the abundance of literature on this topic, it is unclear why authors choose not to tackle it. 

Minor comments:

1) line 134 - "5* of the GR promoter" should be 5' of the GR promoter

2) line 170 - "vide section 3.1" should probably be see or view section

3) line 175 - "Adding the plasmid vector" should be further adapted, since this paragraph refers to overexpression of WT gene which was integrated into genome of Reh and Jukart cells by infecting them with lentivirus (therefore transducing them). Pease correct accordingly.

4) line 640 - letter h should be added next to the number 48

Round 2

Reviewer 2 Report

Authors have answered all my major comments. Current version of the manuscript gives solid overview of the mechanisms of steroid resistance in ALL.